# Response of Lawn Grasses to Salinity Stress and Protective Potassium Effect

**Monika Kozłowska** [1], **Hanna Bandurska** [1,*] and **Włodzimierz Breś** [2]

1 Department of Plant Physiology, Poznań University of Life Sciences, Wołyńska 35, 60-965 Poznań, Poland; monika.kozlowska@up.poznan.pl
2 Department of Plant Nutrition, Poznań University of Life Sciences, Zgorzelecka 4, 60-198 Poznań, Poland; wlodzimierz.bres@up.poznan.pl
* Correspondence: hanna.bandurska@up.poznan.pl

**Abstract:** The salinity effects on lawn grasses caused by mine salts (halite and carnallitite) due to road de-icing processes was the aim of this study. Biometric and physiological parameters were evaluated after salt dosage of 50 and 100 g m$^{-2}$ applied to a lawn surface twice and four times, in weekly intervals. The alleviating effect to the salinity on the grasses from potassium enriched soil was also evaluated. Protective effect of potassium included mostly plasma membrane integrity and an increase in the level of photosynthetic pigments. This probably resulted in more efficient photosynthesis and thus increased lawn growth. Simultaneously, only a slight reduction in relative water content (RWC) was noted, so the recorded increase in proline level may indicate its participation in osmotic adjustment. Our results confirm the importance of proper, and even over-optimal, potassium fertilization of lawn grasses exposed to salinity. Moreover, it is advisable to use other fossil salts instead of halite for the de-icing of near-green areas. The mined salt carnallitite which, besides NaCl, contains about 30% of carnalite (KCl·MgCl$_2$·6H$_2$O) could be such a substance.

**Keywords:** elements; lawns; membrane integrity; mine salts; photosynthetic pigments; proline; RWC





## 1. Introduction

Salinity is one of the most serious environmental constraints affecting plant growth and quality. It is responsible for the inhibition of leaf expansion and photosynthesis restriction, leading to a reduction of biomass accumulation [1]. Plant resistance to salinity depends on the duration and severity of stress, as well as on genetic characteristics responsible for its ability to cope with stress.

Glycophytes, which are found in most utility plants, cannot grow in the presence of high salt levels, however in lower saline conditions they do activate diverse resistance mechanisms [2]. Soil salinity is a significant constraint which cause growth restriction of forage grasses and cereals [3]. Turfgrass species were defined as salt sensitive, moderately sensitive or only moderately resistant plants [4,5]. Nevertheless, lawn cultivars of turfgrass species are valuable and suitable for urban areas thanks to their growth intensity and durability. However, in the northern hemisphere urban areas are often exposed to road de-icing salts of which halite is the most commonly used, containing nearly 98% sodium chloride [6,7]. Excessive soil salinity, along with other anthropogenic pollutants, markedly affect the lawns located near salted roads, walking paths and other green areas. In extreme cases, the quality of lawns may deteriorate and even result in death. Negative effects of salinity on the state of a lawn depend on the cultivars of turfgrass species in grasses mixture, as well as on the type and dose of salt used for de-icing. Moreover, weather conditions, limited rainfall and environmental pollution or low-quality of irrigation water can have an influence on the harmfulness of salt [8,9]. Salinity causes ionic stress due to the accumulation of Na$^+$ and Cl$^-$ in plant tissue, which has a deleterious effect on enzyme activity and cell membrane structure and DNA [10,11]. The harmful effect of ionic

stress includes the degradation of chloroplast pigments, which leads to the reduction of photosynthetic rate [12,13].

It was shown that salt stressed plants have a larger internal potassium requirement and increased potassium supply may play an important role in the alleviation of negative effects of salinity [14,15]. High salinity led to potassium deficiency, however an increased potassium dosage resulted in the reduction of adverse effects of stress on physiological activity and plant growth [1,16]. This element regulates plant water management and stomatal behavior, responsible for the maintenance of photosynthetic $CO_2$ fixation, as well as protects chloroplasts from oxidative damages [17,18]. Literature data shows that genetic variations in salinity resistance correlate well with $K^+$ tissue content. High salt sensitive *Arabidopsis* and tomato mutants were characterized by extremely poor capacity to $K^+$ uptake [19,20]. However, salt stress resistant *A. thaliana* genotypes showed a higher expression of $K^+$ transporter genes and $K^+$ content [21]. On the other hand, it was commonly observed that a high level of $Na^+$ in soil solution affects the reduction of $K^+$ uptake, $K^+/Na^+$ ratio and limitation of photosynthetic activity [12]. Thus, the improvement of potassium nutritional status in plants by the increase of $K^+$ supply may result in the maintenance of high cellular $K^+/Na^+$ ratio which is important for plant growth and salt resistance [22].

Plants have developed varied mechanisms responsible for both avoidance and tolerance to stress in cells [23,24]. A well-known response is the accumulation of amino acid proline which plays an important role in the alleviation of negative effects, both osmotic and ionic stress. This multifunctional amino acid is responsible for osmotic adjustment and acts as an enzyme protectant, toxic oxygen derivatives scavenger and stabilizer of subcellular structure [25–27]. Moreover, it serves as an organic nitrogen reserve during stress recovery [28,29].

There is a lot of data focusing on diverse aspects of plant response and resistance to salinity. However, most of it is based on experiments carried out under fully controlled conditions, in hydroponic or perlite systems and with chemically pure NaCl. The aim of this study was to evaluate the naturally occurring salts used for road de-icing, i.e., mine rock salts, halite and carnallitite which contain hydrated potassium and magnesium chloride (carnallite). The effects of these salts on the response of the widely used lawn grasses mixture 'Wemblay', with different dosages and stress duration, were examined. The hypothesis was that carnallitite may have less harmful effects on lawn grasses than halite and could be recommended as de-icing salt particularly for green belts in parks and home gardens. Moreover, the study also evaluated the effect of enriched potassium on the alleviation of halite harmfulness and lawn grass salinity resistance. In order to do that, growth parameters, along with certain physiological measurements, were evaluated.

## 2. Materials and Methods

### 2.1. Grass Cultivation and Treatments

Box-experiments were carried out in a glasshouse, using the commercial lawn grass mixture 'Wembley' (Kemi-Plast Company of Poznań, Poznań, Poland) containing the following species: *Festuca rubra* L. 'Areta'—25%, *Festuca rubra* L. 'Litango'—18%, *Festuca arundinacea* Schreb. 'Aziza'—10%, *Lolium perenne* L. 'Bokser'—20%, *Lolium perenne* L. 'Dancer'—20% and *Poa pratensis* L. 'Evora'—7%. Grasses were cultivated in plastic openwork boxes 36 cm $\times$ 28 cm, lined with agro-textile and filled with 10 $dm^3$ of sandy soil and peat mixture (2:1), compacted to 8.5 cm in height. Content of NPK was supplemented, providing the following final levels: N-$NO_3$—120, P—80, K—200 mg $\cdot$ $dm^{-3}$ using $NH_4NO_3$, $KH_2PO_4$ and $KNO_3$ [30]. The $pH_{H2O}$ of grown medium was 6.5.

Salts came from Kłodawa Salt Mine SA (central Poland); halite contained 95% of NaCl with evaporate deposit of sulfates, halides and borates and carnallitite contained nearly 70% of NaCl and 30% of carnallite ($KCl \cdot MgCl_2 \cdot 6H_2O$), both with a slight admixture of unidentified minerals [31] and Kłodawa Salt Mine data). Experiments were carried out in the spring (end of March–June) during a two-year period and some were repeated in a third year. Plants were grown under partially regulated conditions; natural light with shades

and air temperatures 15–25 °C. Distilled water was used for irrigation. Water content in the soil was kept at the level of 60% capillary water capacity.

In this case, 3 weeks after sowing, first grass cuttings 5 cm above the soil were performed and subsequent cuttings 2–3 cm above the soil were repeated every 7 days until the end of the experiment. After four cuttings, salt treatment was started, using 300 cm$^3$ of salt solution per 10 dm$^2$ of experimental surface in one plastic box. The concentrations of salts were selected according to the guidelines of the Polish General Directorate of National Roads and Motorways [32]. Salts were applied at weekly intervals two or four times, in 2 different experiments. Experiment 1 included halite and carnallitite dosages: $4 \times 50$, $2 \times 100$ and $4 \times 100$ g $\cdot$ m$^{-2}$ and control without salt, at basic potassium level. Hence, the experimental system was comprised of 28 openwork boxes [(control + two salts × three salinity levels)] × four replicates. Experiment 2 included potassium supplementation at basic (i.e., 200 mg $\cdot$ dm$^{-3}$) level recommended by Kleiber et al. [30], and enriched (400 mg $\cdot$ dm$^{-3}$) level with the following halite dosages: $2 \times 50$ and $2 \times 100$ g $\cdot$ m$^{-2}$ and controls without salts. Therefore, the experimental system was comprised of 24 boxes (2 potassium levels × 3 salinity levels × 4 replicates). Samples of grasses were hand cut and collected for the biometric and physiological analyses. The evaluations of Experiment 1 were performed 7 and 28 days after the last salt application while evaluations of Experiment 2 were performed 10 and 20 days after the last salt application.

Fresh and dry matter of cut grass, intensity of growth (cm/week), leaf relative water content, ions leakage, chloroplast pigments (chlorophyll a and b, carotenoids) and proline content were determined. Dry matter was calculated as percent of fresh mass, following the drying for 48 h at 80 °C. At the end of the experiments the content of minerals in cut grasses was also estimated.

### 2.2. Physiological Parameters

Water content and ions leakage were estimated immediately after grass harvesting. Plant material for the assessment of other parameters was frozen in liquid nitrogen and stored at −20 °C until analyses. Each replicate derived from a separate sample of randomly chosen cut grasses from one openwork box.

### 2.3. Relative Water Content (RWC)

RWC was measured according to the standard method developed by [33], as described by [34] and it was calculated using the following formula:

$$RWC = \frac{fresh\ matter\ -\ dry\ matter}{resh\ matter\ at\ full\ turgor\ -\ dry\ mate} \cdot 100 \tag{1}$$

### 2.4. Electrolyte Leakage (EL)

Electrical conductivity (EC) was measured using the [35] method to evaluate membrane permeability. A dozen leaves cut into 2 cm segments were washed three times in deionized water to remove surface contaminants. Then leaf pieces were put into a 50 cm$^3$ flask, submerged in 10 cm$^3$ of deionized water and kept for 24 h at 10 °C. After warming to 25 °C and shaking, the EC1 of bathing solution was measured. Next, tissues were killed by autoclaving for 15 min, cooled down to 25 °C and EC was measured once again (EC2). Electrolyte leakage was defined as EC1/EC2 and was expressed in percentage.

### 2.5. Proline

Proline content was determined according to [36] with some modifications [34]. Plant material (100 mg) was homogenized with 4 cm$^3$ of 5% TCA and centrifuged at $5000 \times g$ for 15 min. The level of proline was determined by spectrophotometry (at 520 nm) of the quantity of colored reaction product of proline with ninhydric acid. Its amount was calculated from a standard curve and expressed in milligrams per gram of dry mass (mg $\cdot$ g$^{-1}$ DM).

### 2.6. Chloroplast Pigments

Pigments were determined according to [37]. Leaf samples (100 mg) were cut into 2 cm pieces, treated with 5 cm$^3$ dimethyl sulphoxide (DMSO) and incubated for 60 min in a water bath at 65 °C. Optical density of extracts was measured at 663, 645 and 480 nm. The content of chlorophyll a, chlorophyll b and carotenoids were calculated following the modified Arnon formulae [38] and given in milligrams per gram of dry mass (mg · g$^{-1}$ DM).

### 2.7. Mineral Analysis

Leaf samples taken from the last cutting of cut grasses were used for the estimation of the content of N, P, K, Ca, Mg, Na and Cl. Fresh matter was pre-dried at 105 °C for 48 h, ground in a mixer and mineralized with a mixture of $H_2SO_4$ and $H_2O_2$ (2:1). The following analyses were used: N—Kjeldahl procedure; K, Mg, Ca and Na—atomic absorption spectroscopy (AAS); P—spectrophotometric method [39]. For chlorine determination, plant material was mineralized at 500 °C. The residue was dissolved in hot deionized water and after sedimentation the content of Cl was determined by nephelometric method [40]. Results are expressed as a percentage of dry matter (% DM).

### 2.8. Statistical Analysis

All numerical analyses were performed using the STATISTICA 13.3 package (StatSoft, Inc. Tulsa, OK, USA). Following the analyses of variance (ANOVA) with repeated measurements (*n* = 4), Duncan's multiple range post hoc test (for biometric parameters and ions content) and Tukey's post hoc test (for physiological parameters) were used. Differences between means of treatment were compared at significance level at α = 0.05. The differences were considered statistically significant for *p* less than or equal to α.

## 3. Results

### 3.1. Experiment 1—Effects of Mine Rock Salts: Halite and Carnallitite on Salinity Stress

Statistically significant effect of salt type and dose, as well as the interaction between salt type and dose on fresh mass and growth, was found (Table 1). Significant effect of salt type and dose on RWC was revealed after 7 days of salt application. Whereas, after 28 days of salt application, a significant effect of salt dose and interaction between salt type and dose on RWC was found.

**Table 1.** ANOVA results for the effect of salt type and dose on fresh mass, growth and RWC.

| I Term (7 Days after Treatment) | df | Fresh Mass | | Growth | | RWC | |
|---|---|---|---|---|---|---|---|
| | | F | *p* | F | *p* | F | *p* |
| Salt type | 1 | 129.91 | 0.0000 | 2.59 | 0.1209 | 11.84 | 0.0021 |
| Dose | 3 | 105.98 | 0.0000 | 103.83 | 0.0000 | 31.36 | 0.0000 |
| Salt × dose | 3 | 14.50 | 0.0000 | 0.65 | 0.5876 | 1.42 | 0.2602 |
| II term (28 Days after Treatment) | df | Fresh Mass | | Growth | | RWC | |
| | | F | *p* | F | *p* | F | *p* |
| Salt type | 1 | 37.68 | 0.0000 | 66.82 | 0.0000 | 1.49 | 0.2338 |
| Dose | 3 | 83.68 | 0.0000 | 203.11 | 0.0000 | 12.70 | 0.0000 |
| Salt × dose | 3 | 7.85 | 0.0000 | 13.19 | 0.0000 | 6.40 | 0.0025 |

Both salts caused a reduction of fresh mass and growth intensity of lawn grasses (Table 2). This effect was observed at each of used dosages, excluding the lowest carnallitite one. However, the effect of halite was much stronger than that of carnallitite. On the first term of determinations (7 days after the last salt treatment) a reduction of less than 50% of fresh mass was noted at the lowest halite dosage (50 g · m$^{-2}$). Both higher halite dosages caused a 3–4-fold decrease of fresh mass and a reduction of about 40% of growth intensity. At those dosages, carnallitite caused a reduction of about 30–40% in both fresh mass and growth intensity. A slightly weaker effect was observed 28 days after the last salt treatment. The reduction of fresh mass was at about 40–70% and of growth intensity

at about 20–50% at the two higher halite dosages. Whereas only the highest carnallitite level ($4 \times 100 \cdot$ g m$^{-2}$) caused a reduction of about 50% and 30% of fresh mass and growth intensity, respectively.

**Table 2.** Salinity effect induced by halite and carnallitite on fresh mass and growth intensity of lawn grasses.

| Treatments | Dose of Salt (g · m$^{-2}$) | Fresh Mass (g · m$^{-2}$) | | Growth (cm · week$^{-1}$) | |
|---|---|---|---|---|---|
| | | I | II | I | II |
| control | 0 | 201.9 a | 215.5 a | 10.75 a | 10.25 a |
| | 4 × 50 | 115.9 b | 182.0 ab | 9.00 b | 9.00 ab |
| halite | 2 × 100 | 57.6 c | 127.0 b | 6.75 cd | 8.50 b |
| | 4 × 100 | 44.6 c | 65.8 c | 6.25 d | 5.00 d |
| | 4 × 50 | 199.9 a | 205.5 a | 9.25 b | 10.70 a |
| carnallitite | 2 × 100 | 139.2 b | 206.7 a | 7.50 c | 9.25 ab |
| | 4 × 100 | 121.8 b | 111.2 b | 6.50 d | 7.00 c |

I and II—7 and 28 days after the last salt treatment, respectively. Means within each column followed by the same letter are not significantly different at $p \leq 0.05$.

Changes in RWC induced by both salts were significantly weaker compared to the growth changes (Figure 1). Statistically significant decrease in leaf RWC was observed under the highest salinity level ($4 \times 100$ g · m$^{-2}$) of both halite and carnallitite, on the first term of determinations (7 days after salt treatment). A similar RWC decline was also induced by carnallitite after 28 days of salt stress duration.

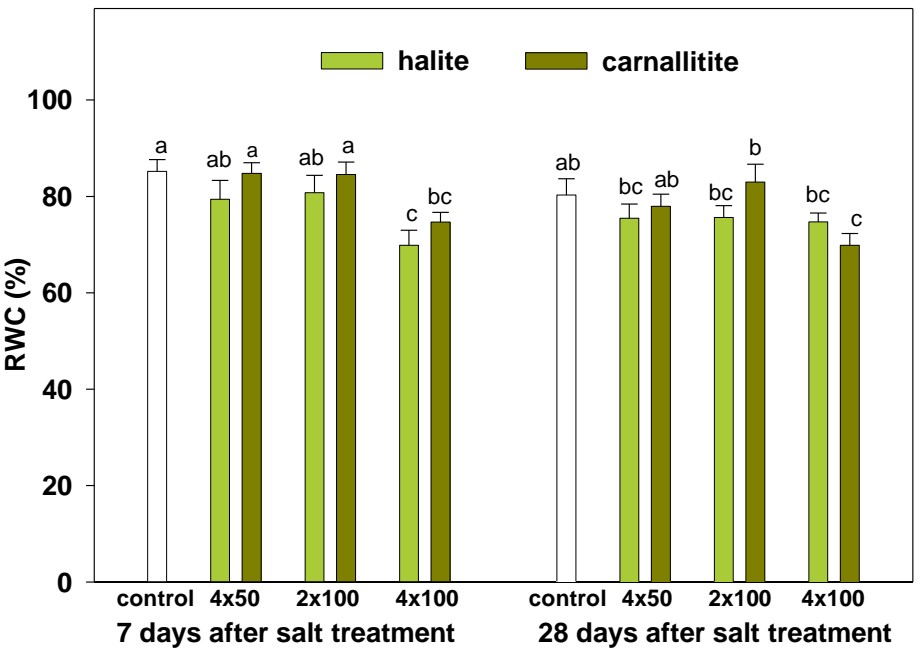

**Figure 1.** Relative water content (RWC) of lawn grasses exposed to salinity stress induced by three doses of halite and carnallitite ($4 \times 50$, $2 \times 100$ and $4 \times 100$ g · m$^{-2}$), estimated 7 and 28 days after the last salt treatment. Data are a means ± SD marked as vertical bars. Means within each term (7 and 28 days) followed by the same letter are not significantly different at $p \leq 0.05$.

*3.2. Experiment 2—Effects of Enriched Potassium on Salinity Stress Response*

The effect of enriched potassium level on growth and physiological response of lawn grasses cultivated under two dosages of halite was examined. The measurements were performed 10 days after the second salt treatment, and after the next cutting, i.e., 20 days of stress duration. Both K level and salt dose significantly affected biomass accumulation.

However, the interaction between K level and the salt dose was statistically insignificant (Table 3).

**Table 3.** ANOVA results for the effect of potassium level and salt dose on fresh mass and dry mass.

| Source of Variation I Term (10 Days after Treatment) | K Level (df 1) | | Salt Dose (df 1) | | K Level × Salt Dose (df 1) | |
|---|---|---|---|---|---|---|
| | F | *p* | F | *p* | F | *p* |
| Fresh mass | 147.63 | 0.0000 | 56.92 | 0.0000 | 1.55 | 0.2369 |
| Dry mass | 187.71 | 0.0000 | 45.32 | 0.0000 | 0.13 | 0.7284 |
| II Term (20 Days after Treatment) | K Level (df 1) | | Salt Dose (df 3) | | K Level × Salt Dose (df 3) | |
| | F | *p* | F | *p* | F | *p* |
| Fresh mass | 13.78 | 0.0029 | 217.23 | 0.0000 | 2.94 | 0.1122 |
| Dry mass | 6.87 | 0.2230 | 113.66 | 0.0000 | 5.88 | 0.0321 |

Double content of potassium in the soil affected the improvement of growth, especially in the earlier period of stress (Table 4). Fresh and dry mass of cut grasses grown with both halite dosages were almost 2.5- and 5-fold higher compared to the basic potassium level. Whereas, at the later date of stress, the difference between basic and enriched potassium level was much smaller. Both growth parameters reached a higher level of about 80% at enriched potassium, but only at the higher dosage of halite.

**Table 4.** Comparison of the effect of potassium levels—basic (200 mg $\cdot$ dm$^{-3}$) and enriched (400 mg dm$^{-3}$) on fresh and dry mass of lawn grasses, exposed to salinity stress.

| Potassium Level | Dose of Salt (g $\cdot$ m$^{-2}$) | Fresh Mass (g $\cdot$ m$^{-2}$) | | Growth (cm $\cdot$ week$^{-1}$) | |
|---|---|---|---|---|---|
| | | I | II | I | II |
| basic | 2 × 50 | 70.0 c | 149.2 a | 13.8 c | 27.8 a |
| | 2 × 100 | 21.6 d | 47.4 c | 4.8 d | 11.2 c |
| enriched | 2 × 50 | 173.0 a | 161.6 a | 31.6 a | 28.0 a |
| | 2 × 100 | 105.4 b | 81.0 b | 22.6 b | 17.6 b |

I and II—10 and 20 days after the last salt treatment, respectively. Means within each column followed by the same letter are not significantly different at $p \leq 0.05$.

ANOVA results showed that halite dose did not have an effect on the content of N and Mg in cut grasses (Table 5). Enriched potassium level showed statistically significant effect on the content of K and Na. Both salt dose and potassium revealed statistically significant effect on K/Na ratio. However, statistically significant interaction between K level and salt dose on the level was found only for the Na content.

**Table 5.** ANOVA results for the effect of potassium level and salt dose on mineral elements content.

| Source of Variation | K Level (df 1) | | Salt Dose (df 1) | | K Level × Salt Dose (df 1) | |
|---|---|---|---|---|---|---|
| | F | *p* | F | *p* | F | *p* |
| N | 2.21 | 0.1625 | 3.81 | 0.0748 | 2.12 | 0.1713 |
| P | 0.55 | 0.4736 | 11.58 | 0.0052 | 0.55 | 0.4736 |
| K | 44.46 | 0.0000 | 20.70 | 0.0007 | 2.22 | 0.1619 |
| Mg | 0.94 | 0.3516 | 10.44 | 0.0072 | 0.42 | 0.5304 |
| Ca | 1.11 | 0.3129 | 20.84 | 0.0007 | 0.67 | 0.4286 |
| Na | 41.75 | 0.0000 | 1526.53 | 0.0000 | 9.78 | 0.0087 |
| Cl | 3.91 | 0.0714 | 284.55 | 0.0000 | 0.55 | 0.4727 |
| K/Na | 17.84 | 0.0012 | 347.01 | 0.0000 | 3.48 | 0.0866 |

At the end of the experiment (20 days after the last salt treatment) Na and Cl content was significantly higher with the higher dose of halite (Table 6). The level of P, Ca and Mg was slightly lower with higher dose of salt. In plants grown with enriched potassium the content of this element increased but the content of Na decreased in grass tissue. Enriched potassium level showed a favorable effect on the K/Na ratio, especially in plants grown with lower halite dose.

**Table 6.** Mineral elements content and the K/Na ratio in lawn grasses grown at basic (200 mg $\cdot$ dm$^{-3}$) and enriched (400 mg $\cdot$ dm$^{-3}$) potassium level, exposed to salinity stress, estimated at the end of experiment, i.e., 20 days after the last salt treatment.

| Potassium Level | Dose of Halite (g $\cdot$ m$^{-2}$) | Mineral Elements (% of Dry Mass) | | | | | | | K/Na |
|---|---|---|---|---|---|---|---|---|---|
| | | N | P | K | Ca | Mg | Na | Cl | |
| basic | 2 × 50 | 3.33 a | 0.46 a | 3.45 bc | 0.67 a | 0.47 a | 1.61 c | 4.73 b | 2.14 b |
| | 2 × 100 | 3.29 a | 0.44 ab | 3.35 c | 0.71 b | 0.43 ab | 3.89 a | 6.16 a | 0.86 c |
| enriched | 2 × 50 | 3.55 a | 0.46 a | 3.71 a | 0.67 a | 0.47 a | 1.43 d | 4.63 b | 2.60 a |
| | 2 × 100 | 3.29 a | 0.43 b | 3.51 b | 0.73 b | 0.41 b | 3.37 b | 5.94 a | 1.04 c |

Means followed by the same letter are not significantly different at $p \leq 0.05$.

ANOVA results show that salt dose had a statistically significant effect on all estimated physiological parameters in both terms (Table 7). Futhermore, K level significantly affected carotenoids content also after 10 and 20 days of treatment with halite. However, it affected proline content only after 10 days, and EL, chl a and chl b content only after 20 days. In turn, the significant interaction between K level and salt dose was shown in both after 10 and 20 days for proline content and only after 20 days for EL and chl a content.

**Table 7.** ANOVA results for the effect of potassium level and salt dose on RWC, EL, proline and chloroplast pigments content.

| Source of Variation I Term (10 Days after Treatment) | K Level (df 1) | | Salt Dose (df 2) | | K Level × Salt Dose (df 2) | |
|---|---|---|---|---|---|---|
| | F | p | F | p | F | p |
| RWC | 0.13 | 0.7218 | 6.87 | 0.0060 | 0.94 | 0.4101 |
| EL | 1.74 | 0.2033 | 351.03 | 0.0000 | 3.65 | 0.0463 |
| proline | 45.56 | 0.0000 | 92.16 | 0.0000 | 11.67 | 0.0006 |
| chl a | 2.09 | 0.1651 | 18.57 | 0.0000 | 2.57 | 0.1040 |
| chl b | 0.36 | 0.5568 | 7.54 | 0.0041 | 0.66 | 0.5313 |
| carotenoids | 18.89 | 0.0004 | 21.43 | 0.0000 | 1.55 | 0.2402 |
| **II Term (20 Days after Treatment)** | **K Level (df 1)** | | **Salt Dose (df 2)** | | **K Level × Salt Dose (df 2)** | |
| | F | p | F | p | F | p |
| RWC | 4.33 | 0.0520 | 3.73 | 0.0441 | 1.34 | 0.2880 |
| EL | 283.01 | 0.0000 | 1135.25 | 0.0000 | 346.35 | 0.0000 |
| proline | 3.98 | 0.0615 | 61.70 | 0.0000 | 5.23 | 0.0162 |
| chl a | 19.35 | 0.0003 | 72.04 | 0.0000 | 3.84 | 0.0407 |
| chl b | 9.84 | 0.0006 | 50.04 | 0.0000 | 1.90 | 0.1785 |
| carotenoids | 8.34 | 0.0098 | 41.50 | 0.0000 | 1.42 | 0.2674 |

Water status measured as RWC decreased in both terms only in plants treated with higher salinity dose and was not affected by enriched K level (Figure 2A). However, membrane permeability increased in plants exposed to salinity stress (Figure 2B). The outflow of electrolytes got significantly higher with the increase of salinity. Enriched potassium level affected a 2.5-fold reduction of electrolytes outflow in plants grown at high salinity level (2 × 100 g $\cdot$ m$^{-2}$), but only on the second term.

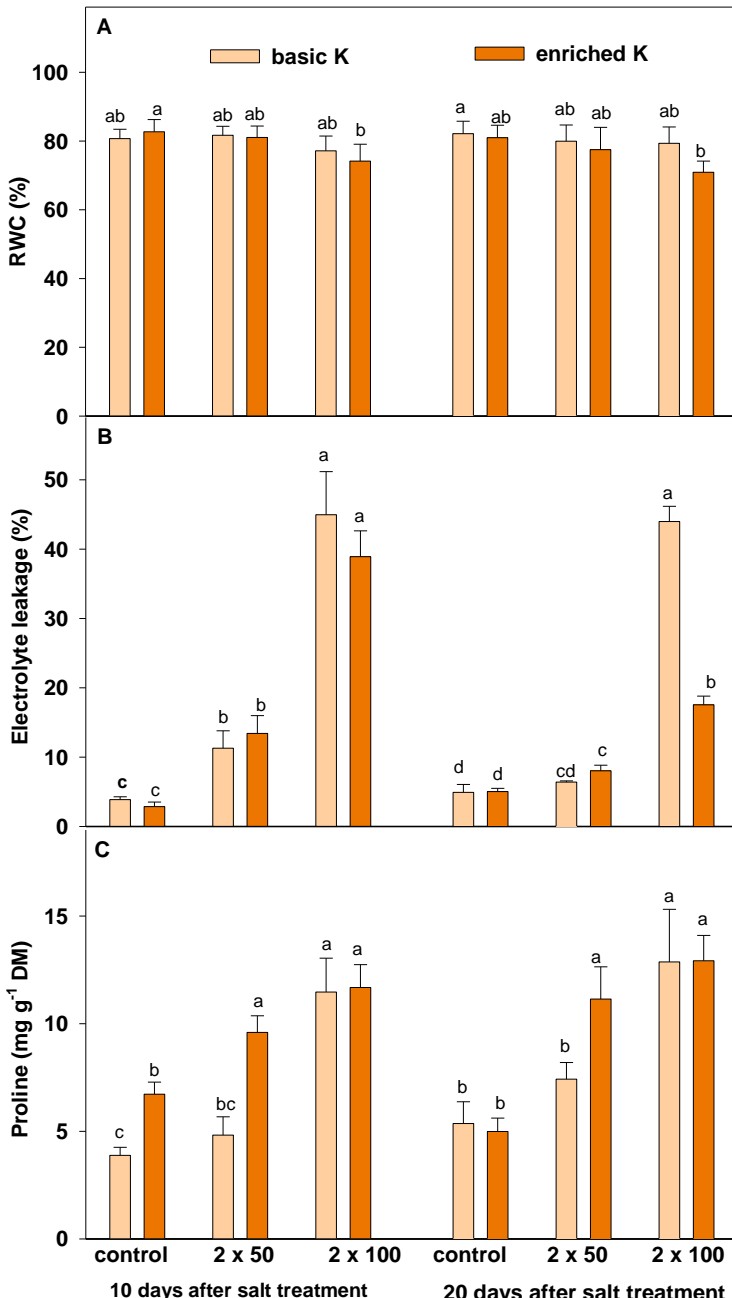

**Figure 2.** RWC (**A**), EL (**B**) and proline content (**C**) of lawn grasses grown under basic (200 mg · dm$^{-3}$) and enriched (400 mg · dm$^{-3}$) potassium level, exposed to salinity stress (2 × 50 and 2 × 100 g · m$^{-2}$), estimated 10 and 20 days after the last salt treatment. Data are a means ± SD marked as vertical bars. Means within each term (10 and 20 days) followed by the same letter are not significantly different at $p \leq 0.05$.

Salinity induced proline accumulation in grass tissue (Figure 2C), wherein the level of this amino acid grew with increasing salinity. At lower halite dosage, proline increased by 20% to 60%, while at higher dosage even more than 2-fold, as compared to control. It has also been shown that enriched potassium level caused an increase of proline accumulation, but only at lower salinity level; its content achieved the accumulation similar to the highest halite effect.

Content of chloroplast pigments was subjected to almost similar changes under the salinity stress, as well as potassium supplementation (Figure 3A–C). In a shorter period of saline treatment (10 days after salt treatment), the higher salt dosage had a degrading

effect on chlorophyll a (Figure 3A) and carotenoids (Figure 3C). This negative effect was mitigated by the increase of potassium level in soil. Salt stress did not change the level of chlorophyll b in the first term. In the second date of stress, the level of all pigments increased in grasses treated with the lower dosage but did not change at the higher salt dosage. Enriched potassium level resulted in a significant increase of chlorophyll a and b in plants grown at the higher dosage of halite.

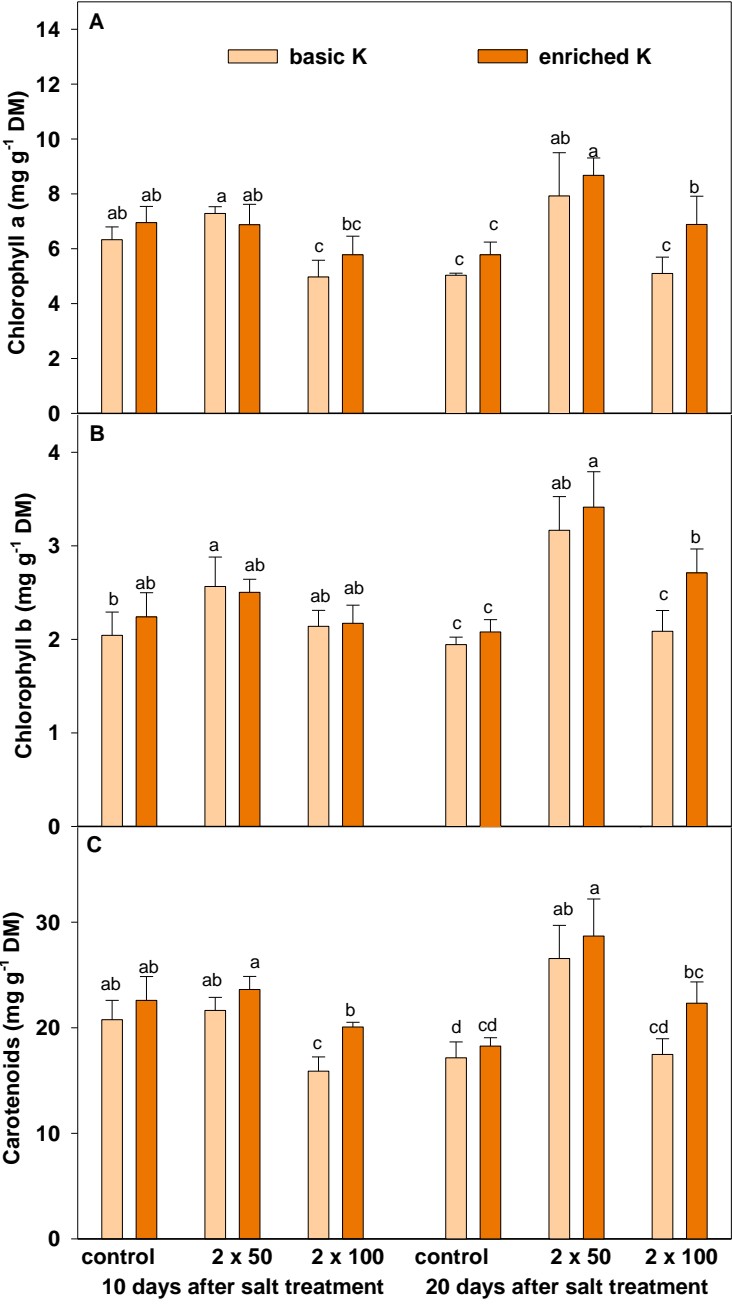

**Figure 3.** Chlorophyll a (**A**), chlorophyll b (**B**) and carotenoids (**C**) content of lawn grasses grown under basic (200 mg $\cdot$ dm$^{-3}$) and enriched (400 mg $\cdot$ dm$^{-3}$) potassium level, exposed to salinity stress (2 $\times$ 50 and 2 $\times$ 100 g $\cdot$ m$^{-2}$), estimated 10 and 20 days after the last salt treatment. Data are a means $\pm$ SD marked as vertical bars. Means within each term (10 and 20 days) followed by the same letter are not significantly different at $p \leq 0.05$.



## 4. Discussion

Millions of hectares of land around the world are affected by soil salinity. Generally, it results from excess NaCl accumulation, causing osmotic stress and from an accumulation of toxic ions. Salinity has a harmful effect on water relation, photosynthesis= and other physiological processes [24,41].

Salt stress, as a result of the use of road de-icing salts, limits the development of lawn grasses and is a nuisance for people. According to Polish regulations, which are an answer to weather conditions, a single non-toxic application of NaCl should not exceed 30 g m$^{-2}$ but the total seasonal dose of salt usually ranges from 1350 to 1600 g · m$^{-2}$ [32]. In our studies, using rock salts and potassium as an essential nutrient and protective substance, single salt dose was higher, i.e., 50 and 100 g · m$^{-2}$. However, the total amount of salt applied during the several-week long experiment was smaller. It ranged from 200 to 400 g · m$^{-2}$ in the halite/carnallitite experiment and 100 and 200 g · m$^{-2}$ in the halite experiment with enriched potassium level.

Both salts reduced lawn growth intensity but did not cause their dieback. Carnallitite salt definitely had a less damaging effect, which indicates a possibility for its de-icing use, especially on roads/pavements surrounded by small green areas, gardens and parks. This is also the intention of Kłodawa Salt Mine SA, using carnallitite deposit to produce relatively low-priced de-icing commercial salts. On the other hand, the harmful effect of halite decreased over time, but was still greater than for carnallitite. The unpublished data shows that the strong toxicity of halite, which leads to grass dieback, occurred only at a single 200-g dosage, hence such dosages cannot be used in the de-icing procedure.

The examined grass is a mixture (Wembley) designed for intensively used lawns. It is composed of various species with different salt stress resistance, from sensitive to moderately resistant. *F. rubra* and *P. pratensis* are found to be sensitive to salt stress, nonetheless many of *F. arundinacea* cultivars were estimated as resistant. However, the resistance of *L. perenne* varieties, a common and useful lawn grass, has not been observed [4,7,42]. It should be noted that a multiplicity of species and cultivars makes it possible to create a turfgrass mixture with specifically optimized functional features, including salt resistance.

The most detrimental effects of salinity stress were a result of the accumulation of Na$^+$ and Cl$^-$ ions in plant tissues. They caused severe ion imbalance and excess uptake which may result in physiological disorders, leading to a significant reduction of aboveground mass and lowered quality of lawn grasses (Table 1). In the case of carnallitite salt, the content of harmful Na$^+$ and Cl$^-$ ions was about 1/3 lower, and furthermore, potassium and magnesium ions from carnallite were more available.

The involvement of potassium in resistance to salinity is widely documented and reviewed in literature [17,22,43]. Low harmfulness of carnallitite prompted us to investigate the potentially beneficial protective effect of potassium to salt stress. Our results have shown that in plants growing under enriched potassium level, the content of this element was slightly higher only at lower salinity. However, it should be noted that the estimation of elements content was made at the end of the experiment. High Na$^+$ concentration in salt stressed plants affects the inhibition of K$^+$ ions uptake and, therefore, a deficiency of K$^+$ could be observed [44,45]. Moreover, due to the similarity between these ions, voltage-dependent K$^+$ channels appear to be one of the pathways for the entry of Na$^+$ ions into the plants [12,46]. In our experiment the content of both Na$^+$ and Cl$^-$ doubled with growing salt dosage but this increase was slightly lower in plants grown with enriched potassium level and K$^+$/Na$^+$ ratio was also more favorable (Table 3).

It can be assumed that enriched potassium dosage affected the minor reduction of fresh mass than the one observed at the basic potassium fertilization. (Table 2). Plants exposed to salinity are characterized by greater potassium requirements [22]. As an important macro-nutrient, it is essential for cell expansion, regulation of water relation and photosynthesis. Under conditions of salt stress, Na$^+$ competes with K$^+$ for major binding sites in key metabolic processes and for transporters at the plasma membrane. Moreover, it lowers K$^+$ uptake and disturbs its translocation from roots to shoots.

Salinity lowers water potential which may cause an occurrence of water deficit in plant tissues. The presented results show that the applied salt stress caused only a slight reduction of RWC (Figure 1) independent of potassium application (Figure 2A). However, a slight decrease of RWC and high membrane injury (Figure 2B) indicate a direct toxic effect of $Na^+$ and $Cl^-$. The increase in electrolytes leakage connected to plasma membrane damages are the basic effects of salinity [16]. However, potassium involvement in maintaining of plasma membrane integrity was found. Plants grown with enriched potassium level and treated with high halite doses were characterized by only small membrane disintegration, especially at a later date of salt treatment. According to [17,22], the improvement of $K^+$ nutrition under salt stress conditions could be essential for minimizing oxidative stress which has a positive effect on membrane integrity.

The applied higher dosage of halite caused a significant reduction of some chloroplast pigments but only at the first date of the analyses (Figure 3). Similar effects were noted in many plants, including different turf grass species in which chlorophyll a decreased much more than other pigments [43]. Salinity-induced chlorophyll reduction may be caused by chlorophyll oxidation, related to ROS generation and/or Mg deficiency [47]. Furthermore, it was reported that salinity induces the activity of some chlorophyll degrading enzymes and disturbs thylakoid membranes and chloroplast structure [11,48]. As a consequence, the rate of photosynthesis may be reduced and hence plant growth is restricted [12]. The results described here showed that the application of enriched potassium had a positive effect on the reduction of chloroplast pigments degradation. Such a stimulating effect was not expected. However, it was fully confirmed in an independent additional experiment performed in the following year. How does potassium stimulate the build-up of photosynthetic pigments? It is probably not directly involved in the synthesis and enhanced accumulation of chloroplast pigments. Nonetheless, it can enhance its synthesis by an improved uptake of macro- and micro-nutrients and plays an important role in this process [18]. Enhanced potassium level may also affect the lowering of ROS production which protects chloroplasts from oxidative damage and pigments degradation [49].

The multifunctional substance which plays an important role in the alleviation of harmful effects of both osmotic and ionic stress caused by salinity is proline [26,27,50]. High proline accumulation in examined lawn grass treated with higher halite dosage, along with only slight RWC reduction, confirms the involvement of this amino acid in osmotic adjustment responsible for the maintaining of tissue hydration, as was also found in turfgrass species by [43]. Moreover, as ROS scavenger, proline could protect examined lawn grasses against the degradation of chloroplast pigments and PSII damage, which was also shown in other plant species [51,52]. This protective function of proline is particularly important in preventing the effects of ionic stress which occurs later than osmotic stress. Our study indicates that the application of enriched potassium dosage provided a significant increase in proline content. Likewise, the application of potassium contributed to an increased proline accumulation in drought stressed *Nicotiana rustica* L. [15]. However, the mechanism responsible for this effect is unknown. Our research has shown that the content of this amino acid in lawn grasses, growing with higher potassium level and treated with lower halite dosage, was the same as in those treated with higher halite dose. It was particularly important for the protection of cell membranes and chloroplast pigments against ionic stress in the second term.

## 5. Conclusions

The naturally occurring carnallitite is characterized by a significantly lower harmfulness for lawn grasses as road de-icing substance. Vegetative lawn growth was only slightly limited in contrast to commonly used halite.

It seems that this less harmful effect is caused by the presence of potassium in carnallitite salt. Potassium affects the improvement of plant functioning in saline conditions by lessening the accumulation of sodium and chlorine ions, diminishing the degradation of chloroplast pigments and increasing the accumulation of proline. The use of carnallitite as

a less harmful de-icing salt means increased durability of lawns in urban areas as well as makes it possible to cultivate grass species moderately resistant to salinity. However, when using halite potassium fertilization of lawn grasses should be enriched.

**Author Contributions:** Conceptualization and project administration, M.K.; methodology, validation, formal analysis, investigation and resources, M.K., H.B. and W.B.; writing—original draft preparation, M.K.; writing—review and editing, M.K., H.B. and W.B.; visualization, M.K. and H.B.; supervision, M.K. and W.B. All authors have read and agreed to the published version of the manuscript.

**Funding:** Publication was co-financed within the framework of the Polish Ministry of Scienceand Higher Education's program: "Regional Initiative Excellence" in the years 2019–2022 (No.005/RID/2018/19)".

**Institutional Review Board Statement:** Not applicable.

**Informed Consent Statement:** Not applicable.

**Data Availability Statement:** Not applicable.

**Acknowledgments:** The authors would like to thank Magdalena Rybus-Zając and Małgorzata Zielezińska, Renata Matysiak and Ilona Bernat for their participation in analytical research.

**Conflicts of Interest:** The authors declare no conflict of interest.

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
