# Peer review of "Response of Lawn Grasses to Salinity Stress and Protective Potassium Effect"

_agronomy, doi:10.3390/agronomy11050843_

Round 1
Reviewer 1 Report
- The introduction can be largely reduced.
- Major concerns about the experimental design. Line 109 - 115. Hard to follow in reading. E.g. 4 x 50 g m-2. What does '4' mean? Need to clearly describe in M & M. Secondly, how are the concentrations of halite and carnallitite selected? In the second experiment - K supplement, what are the K application rate for the basic and enriched levels? How are they determined? A treatment with no K application should be included. Both experiments seem to be factorial design, are there interactions between salt type x salt level or K level x salt level? The above questions must be addressed.
- In the results, the letter assignment is strange. For example, In Table 1, 'c' represents the highest value, 'a' in the middle, while 'b' represents the lowest value. In Figure 1, foot note, 'Means within each term followed by ...'. What is 'term' referring to? Table 3, where are the numbers for the control treatment? Why can't K/Na be statistically analyzed? For Expt 1. it is more important to compare salt type than salt level. And Expt. 2, it is more important to compare K level than salt level. All tables and figures should be adjusted accordingly.
Reviewer 2 Report
In the current manuscript, the authors signify the vital role of potassium in mitigating salinity stress. Overall, the experiment was nicely planned and executed. The authors concluded that the application of enriched potassium level contributed to improving plant functioning in saline conditions by lessening the accumulation of sodium and chlorine ions, diminishing the degradation of chloroplast pigments and increasing the accumulation of proline. I do not see any significant issue with the study since there are very routine practices unless provided with molecular insight. Mainly, all investigation has been carried out in boxes, which should be measured as a constraint related to this work—a few minor suggestions.
- Line 16 20, 47, 58, define abbreviations. Please check the entire text and define the undefined abbreviations on the first appearance.
- Write the RWC formula using equation editor in the MS office.
- Line 371-376, please make a separate heading for a conclusion. Also, add some future recommendations on how this study can benefit local grass growers.
Reviewer 3 Report
Authors have evaluated in this study the effect of salinity stress and protective potassium in lawn grasses growth. The manuscript presents interesting results in grass but authors should improve some parts in the text. Following, I have included some comments aimed to enhance the paper:
- In material and methods, authors said that : The evaluations were as follows: 7, 14 and 28 days after the last dose of salt application, but in the table 1 they have presented only results of 7 and 28 days.
- Table 1: add the p values or the level of significance.
- Authors reported in line 170-171 that Standard deviations were also calculated and presented in tables and figures (means ± SD). Authors do not showed in Table 1, 2, 3 the standard deviations (in the figure, SD is presented). Authors must add DS in all tables.
- Table 1, can author explained the difference in fresh mass between the treatment of 7 and 28 days, the table showed that with more days of salt application, the fresh mass increased.
- In the Figure 1, it better to add the means values in the histogram and p values.
- In discussion line 293-296, authors indicated shortly the mixtures of grass, but they do not report their effect in the salt resistance. Authors informed in the discussion that: The examined grass is a mixture (Wembley) designed for intensively used lawns. It is composed of various species with different salt stress resistance, from sensitive to moderately resistant. But authors do not discuss anything about the advantages of each species used in the mixture, if Poa or Lollium or fescue provide more resistance to salinity with the same manner, or which of them is the best contributing the salt resistance according to the species characteristics. It is not the same to compare the study of a single grass or a mixture of grasses.
- L 317and its increased amount could be in the roots as well. Authors cannot confirm this, since they have not evaluate roots.
- Finally, the topic of this manuscript in interesting. Authors must added some details of species used in this study, and their advantages to tolerate salinity stress in the discussion to improve the manuscript.
Round 2
Reviewer 1 Report
Experimental design and data analysis are major concerns. Need to be addressed.

Author Response
Thank you once more for valuable comments and suggestions.
All changes and correction have been made according to reviewer’s suggestions and are marked in red.
Rev. 1
Reviewer’s comments and suggestions, and authors’ answers
- Experiment design - Experiment 1 (salt type) is inconsistent with sas analysis. In the M&M, the authors stated that it included 7 treatments (2 salt type x 3 dose + 1 non-salt control) x 4 reps = 28 pots. In the sas table (Table 1), df of dose = 3. Then that would include the non-salt treated control. The design would be 8 treatments (2 salt type x 4 dose) x 4 reps = 32 pots.
There was only one pot for control (plants grown without salt). So, control for each salt (halite, carnallitite) was the same. RWC, as well as growth parameters, was determined only once for the control. The experimental system was comprised of 28 openwork boxes included control, two salts in three salinity levels all in four replicates. So, there were 7 treatments because there was one control box. In our opinion there was no need using two control pots (one for each type of salt). Therefore for the purposes of ANOVA analysis the results for both salts (3 dose) were related to the same control. So, for the first factor (salt dose) the values of determined parameters for the control was the same for each salt type (entered twice in ANOVA calculation table) and therefore df = 3.
- Experiment 2: (1) Why the growth was only evaluated under the saline conditions (df=1), while other parameters (e.g. proline, RWC) were quantified under three saline conditions (df = 2)?
The effect of salt dose on plant growth was shown in Exp. 1 This experiment also shown that less harmful effect on growth inhibition revealed potassium-containing carnallitite.
So, the main purpose of Exp. 2 was to examined the effect of soil supplementation with enriched potassium level on plant growth under saline conditions. So, only two dose of halite was used.
Only physiological parameters which have not been examined in Exp. 1 were tested both under control (without salt) and saline conditions.
- Where is the ANOVA of mineral analysis? Na/K ratio can be calculated with each salinity level and then analyzed statistically. Why only calculated using the mean? It is not helpful to report Na/K without statistical analysis. ANOVA table is needed to show the statistical result, regardless if interactions are observed or not. Need to address these after mentioned major problems.
ANOVA results for minerals were included and statistical analyses for K/Na were also supplemented.
- Change to Experiment 2 instead of 'the second one'. It is much clear using Experiment 1 and Experiment 2 to describe the two experiments in M & M and Results instead of treatment (e.g. halite and enriched potassium).
It was changed according suggestion
- Suggest changing to 'basic (i.e. xxx kg/ha recommended by xxx (citation)). Need to add the K rate in the M & M. Citation 30 should be used here.
In the responding letter, the author stated that the information about K rate was added. Did not say in the revision.
It was changed and related paper [30] was cited. According to developing fertilization recommendations in horticulture [30] we provide information regarding K supplementation (and also N and P) in mg dm-3 which indicates the content of nutrients in the soil. The rate of nutrient supplementation was determined according to Kleiber et al [30] and with regard to the level of these elements in soil used in experiment.
- It should be [(control + two salts x three salinity levels)] x 4 reps. The control should have 4 reps as well.
Yes, it was corrected.
- Suggest changing to 'The evaluations of Experiment 1 were performed 7 and 28 days after.., while the evaluations of Experiment 2 were...... 10 and 20 days..."
It was changed
- Suggest adding 'Experiment 1 - effects of halite/carnallitite .....' as a subtitle to clearly separate the two experiments.
- Add (Tab. 1) here
- Suggest using Day 7 or Day 28. 'Term' is hard for readers to follow.
Yes all of these suggestions have been included
- If the objective is more to compare salt type halite vs. carnallitite, suggesting group the results by salt dose on the x axis. The same suggestion for Experiment 2.
All figures was rearranged according this suggestions.
- Suggest adding Experiment 2 as an subtitle. – It was add
- Where are the data for the non-salt treated control?
In Exp. 2 growth parameters were determined only in saline conditions in plant grown with basic and enriched potassium. The aim of this experiment was to compare the effect of K levels on biomass accumulation under salinity stress conditions.
Additional authors information
We have also found mismatch between tables and figures in the way of marking the differences between means using letters. It has been corrected so that in figures and tables the highest means are marked as a and the next b, c ...
Moreover, language correction was also made.
